# NH_3_ Plasma-Treated Magnesium Doped Zinc Oxide in Biomedical Sensors with Electrolyte–Insulator–Semiconductor (EIS) Structure for Urea and Glucose Applications

**DOI:** 10.3390/nano10030583

**Published:** 2020-03-23

**Authors:** Chun Fu Lin, Chyuan Haur Kao, Chan Yu Lin, Kuan Lin Chen, Yun Hao Lin

**Affiliations:** 1Department of Electronic Engineering, Chang Gung University, Kwei-Shan, Tao-Yuan 333, Taiwan; chiunfu0513@hotmail.com (C.F.L.); d1711956@gmail.com (K.L.C.); dbp0901@gmail.com (Y.H.L.); 2Kidney Research Center, Department of Nephrology, Chang Gung Memorial Hospital, Chang Gung University, Taoyuan City 333, Taiwan; r5234@cgmh.org.tw; 3Department of Electronic Engineering, Ming Chi University of Technology, New Taipei City 24301, Taiwan

**Keywords:** biomedical sensors, Mg-doped ZO (MZO) film, NH_3_ plasma, glucose and urea

## Abstract

This study compared the sensing characteristics of ZnO (ZO) treated with ammonia (NH_3_) plasma for 1 min, 3 min, and 6 min, under the EIS structure. The measurement results revealed that, after 3 min of NH_3_ plasma treatment, the Mg-doped ZnO (MZO) sensing film had a high hydrogen ion sensitivity, linearity, hysteresis, and drift rate of 53.82 mV/pH, 99.04%, 2.52 mV, and 1.75 mV/h, respectively. The sensing film was used with sodium and potassium ion solutions, and it performed satisfactorily in sensing hydrogen ions. Additionally, we investigated the biomedical sensing properties of Mg-doped ZnO (MZO) sensing film with regard to urea, creatinine, and glucose solutions and found that the Mg-doped ZnO (MZO) sensing film treated with NH_3_ plasma for 3 min had the best properties for sensing urea, creatinine, and glucose. Specifically, with glucose, the sensing film achieved the best linearity and sensitivity and of 97.87% and 10.73 mV/mM, respectively. The results revealed that the sensing characteristics varied with the processing environment and are useful in the developing biomedical sensing applications with different sensing elements.

## 1. Introduction

The area of a sensing film is directly immersed into the solution as the pH-ion sensitive field effect transistor (ISFET) operates. The sensing film in contact with the solution to be tested is key for converting the chemical quantity into electrical quantities, using an ion-sensing element. The mechanism responding to the ion activity in a solution is based on the site-binding model proposed by Yates et al. [1] and forms an interface potential at the interface between the electrolyte and the sensing film. Notably, the interface potential varies with the ion activity of the electrolyte solution and modulates the channel conductance of the ISFET, which in turn changes the source current. When the pH-ISFET sensing film is in contact with the solution, an electrical double layer is formed at the interface, to create an interface potential, and its size is related to the sensing film’s material properties and hydrogen ion activity in the solution. As is known from electrochemical theory, the sensing film with a Nernst response is characterized by the abovementioned relationship between the electrolyte potential and the sensing layer’s interface potential, as well as the monovalent ion activity in the solution [2]. The modification of ISFET to the simpler capacitor structure produces Electrolyte–Insulator–Semiconductor (EIS) structure.

To overcome these problems, several metal oxide materials, such as ITO, CeO_2_, and Nb_2_O_5_, have been extensively investigated for bio-sensor applications, owing to their excellent pH sensitivity [3,4,5]. The Zine oxide (ZnO), an n-type semiconductor material with wurtzite structure, has a band gap of (E_g_) 3.3 eV [6]. Additionally, this material is widely used in non-volatile memories, MOSFET, and optical semiconductor devices, owing to its exciting properties. Moreover, because of the absorption and desorption process, it can detect many ion concentrations by varying conductivity of surface [7]. Because pH sensors operate by discovering the volume of hydrogen ions at a buffer solution, ZnO (ZO) can potentially be used for pH sensing, owing to its excellent hydrogen detection ability. MgO has higher energy gap (7.7 eV) in comparison to ZnO (3.37 eV) [8,9,10,11]; MgO helps to widen the energy gap of ZnO (ZO), and the electronegativity of Mg (*X* = 1.31) [12] is lower than that of most ions, which increases its combination with oxygen to form a stable crystalline structure [13]. The vacancies are substituted by the Mg atom, which helps to fix dangling bonds to mitigate the defects in Mg-doped oxides [14]. Additionally, MgO and ZnO reduce the lattice mismatch, as the size of magnesium ions and zinc ions is approximately similar [15]. More electron-hole pairs are generated due to the MZO, which changes the oxygen vacancies. Thereby, it significantly improves the bonding site of ZO nanostructures and may increase the sensing film’s adsorption capacity. This study investigated the effect of Mg is being doped on a ZO film manufactured by using radio frequency (RF) co-sputtering with a target of Mg and ZnO, as well as different NH_3_ plasma treatment times.

To this end, since nitrogen (N) is similar to oxygen (O) in regard to ionic radius and acts as a better compensator, NH_3_ plasma was used in the post-treatment of sputter film of ZO and MZO. Here, we have investigated the impact of NH_3_ plasma treatment on the sensor attribute, surface morphology, and crystal structure of ZO. Additionally, post-growth plasma treatment was carried out, using TEM methods for phase mapping in the MZO treated with NH_3_ plasma, and the effects exerted on the section morphology characteristics of the detector’s film were observed [16]. However, the effects of related chemical reactions, such as densification mechanisms and structural condensation caused by the NH_3_ plasma treatment of solution-treated ZO and MZO films have not yet been elucidated. In addition, during NH_3_ plasma treatment, plasma-induced damage sometimes occurs in localized areas on the surface of the ZO and MZO films. Furthermore, the research has shown that, if the plasma treatment process has long time duration, evident plasma-induced damage appears in the local regions of the ZO and MZO film surface [16]. To investigate the effect of NH_3_ plasma treatment on solution-treated ZO and MZO biosensors, this study incorporated NH_3_ plasma treatment into the preparation of ZO and MZO biosensors, with the time as low as 1–6 min [16]. The activation mechanism of the NH_3_ plasma treatment for the ZO and MZO films was investigated by carrying out exhaustive electrochemistry and physical characterization analyses based on the biosensor characteristics and using X-ray Diffractometry (XRD), Atomic Force Microscopy (AFM), Transmission Electron microscopy (TEM), and Energy Dispersive Spectroscopy (EDS).

## 2. Materials and Methods

To apply ZO to Electrolyte–Insulator–Semiconductor (EIS) structures as the sensing film, fabrication was performed on a 4-inch n-type (100) silicon wafer with a resistivity of 5–10 Ω·cm. To remove the native oxide, the silicon wafer was cleaned using HF (HF:H_2_O = 1:100). Then, as the first condition, a 50 nm ZO sensing film was deposited onto the silicon wafer, using radio frequency (RF) reactive sputtering, under an atmosphere consisting of a mixture of Ar and O_2_ (Ar:O_2_ = 23:2). As the second condition, a 50 nm MZO sensing film was deposited on the n-type silicon wafer by co-sputtering. During the reactive sputtering, ZO was used under an Ar:O_2_ atmosphere of 23:2. The applied RF power was 100 W, and the pressure was 10 mTorr. After deposition, the ZO and MZO were subjected to post-NH_3_ plasma treatment inside a plasma-enhanced chemical vapor deposition (PECVD) system, with an RF power of 30 W and processing pressure of 500 mTorr, for 1 min, 3 min, and 6 min. Then, an Al film with a thickness of 300 nm was deposited onto the backside of the silicon wafer. Next, to define the sensing window, adhesive silicone gel was used. Finally, the samples were fabricated on the copper lines of a printed circuit board (PCB) in silver gel. An epoxy package was used to separate the EIS structure and copper line, as shown in Figure 1.

## 3. Results and Discussion

### 3.1. Physical Characteristics

XRD is a simple, direct, and non-destructive method for analyzing thin films in the early stages of process development. The D-5000 X-ray diffraction made by SIEMENS was used to investigate the crystal structure changes for ZO and MZO films under different parameters. The operating condition was as follows: The X-ray source was a copper target (Cu Kα, λ = 1.54051 Å), the diffraction angle (2θ) was 20–65°, and the scanning rate was 0.025 °/s. The data were compared to information provided by the Joint Committee on Powder Diffraction Standards (JCPDS), to clarify the 2θ value of the lead magnesium and its corresponding lattice planes, as well as to determine the crystal phase of the ZO and MZO film prepared by using different coating parameters, as shown in Figure 2. As can be seen, when the Mg content increased, the crystal structure was still a wurtzite structure. This analysis confirmed the lattice changes of the sensing film in different conditions in the experiment. Figure 2 shows the XRD patterns of the ZO and MZO layer after RF sputtering in different NH_3_ plasma treatments. The crystalline phase of the MZO sample with NH_3_ plasma treatment for 3 min clearly exhibited characteristic XRD peaks. Additionally, three major diffraction peaks of (100), (002), and (101) were observed at 31.75°, 34.50°, and 36.39°, respectively, and they were stronger compared with the other conditions. The reason for this is that the MZO can form a better crystallized phase owing to the stronger bonding formation. The ZO (002) peak shifted from 34.40° to 34.50°. Higher 2θ indicates that smaller Mg^2+^ ions replaced the Zn^2+^ during the co-sputter [9,17,18]. As the atomic size of Mg (0.66 Å) and Zn (0.74 Å) [10] are almost similar, magnesium can be used instead of zinc, without changing the lattice structure. Therefore, compared with Zn, the crystallinity of the thin film improved because the Mg formed stronger bonds with O by forming MgO [19]. Figure 2 shows the two diffraction MgO peaks of (200) and (220) investigated in the MZO film, which could form a better crystalline phase after NH_3_ plasma treatment. Because nitrogen forms a binding bond with the film, a stronger nitride layer exists for bond formation, and this endows the ZO and MZO film with a denser lattice structure. The NH_3_ plasma treatment affects the peak intensity at different times. When the time conditions are 1 min and 6 min, the peak intensity is significantly reduced. Because of NH_3_ dissociation, the nitrogen radicals promote surface passivation. Therefore, NH_3_ plasma treatment is considered to be an effective N_2_ ion plasma treatment for film passivation [20,21,22]. Additionally, the results revealed that an appropriate NH_3_ plasma treatment for 3 min causes an obvious grain increase.

Figure 3 shows the 3D-AFM of sensing films after various NH_3_ plasma treatment periods. To analyze the effect of the ZO and MZO sensing film with different NH_3_ plasma treatment periods using AFM, the root-mean-square roughness value of ZO treated with NH_3_ plasma for 3 min was 1.99 nm. Furthermore, the root-mean-square roughness values of the MZO sensing film treated with NH_3_ plasma for 1 min, 3 min, and 6 min were 1.93, 2.04, and 1.96 nm, respectively. When comparing Mg-doped samples to pure ZO, we observed higher surface roughness in the Mg-doped samples. Because Mg has a lower electronegativity formed MgO, Mg atoms were incorporated to increase the formation crystal grain and improve the sensing performance [18,23].

Due to the NH_3_ plasma treatment, plasma-induced morphological changes and increment of grain size were observed, favoring the increase of surface roughness and number of surface defect sites, and thus resulting in higher sensitivity and linearity [24]. The surface charge density is mainly related to the ionic activity in the solution. The same element acquires different surface site densities in different acid–base solutions, resulting in different surface potentials. Therefore, the relationship between the surface potential and pH can be expressed with Equation (1) [25]:(1)$$2.303\left(pH_{pzc}-pH\right)=\frac{q\varphi_{s}}{KT}+sin^{-1}(\frac{q\varphi_{s}}{KT}\frac{1}{\beta })$$
where pH value with zero surface charge is the $pH_{pzc}$, and coefficient defining the relationship of $\varphi_{S}$/pH is the β, as follows: (2)$$\beta =\frac{2q^{2}N_{s}{\left(K_{b}/K_{a}\right)}^{\frac{1}{2}}}{KTC_{DL}}$$
where surface site density is the $N_{s}$, and electric double layer capacitance value of the Gouy–Chapman–Stern model is the $C_{DL}$. When β is extremely large, the surface is highly responsive; therefore, the sensitivity is better and the overall response is similar to the Nernst response. When β is small, the overall acid–base response is nonlinear, and the sensitivity is poor. Hence, after NH_3_ plasma treatment for 3 min, the MZO exhibits the highest roughness, resulting in higher linearity and sensitivity.

The TEM image of the MZO composite treated with NH_3_ plasma for 3 min is shown in Figure 4. It is clearly evident that the MZO composites have a nanoscale structure. Additionally, ZO exhibited similar columnar structures in the polymerization of nanoparticles. The low-resolution TEM images (LRTEM) (Figure 4a) further confirm this columnar structure and reveal its high density. As shown in Figure 4b, the lattice spacing was approximately 0.15~0.2 nm in the high-resolution TEM image (HRTEM). The uniform composition of the MZO composites treated with NH_3_ plasma for 3 min was further measured by carrying out EDS analysis, as shown in Figure 4c,d. Figure 4d shows a representative elemental mapping of the nitrogen, oxygen, zinc, and magnesium in the composite [26].

### 3.2. Sensing Characterization

The EIS system consisted of a reference electrode, electrolyte, insulating layer, substrate, and metal electrode, as shown in Figure 1c. Additionally, in electrochemistry, the EIS system is typically expressed as follows:Reference electrode∣Electrolyte∣Insulator∣Si∣M

The metal-insulator-semiconductor (MIS) system is formed by replacing the metal gate of the MIS structure with a reference electrode, electrolyte, and sensing film. The flat band voltage is expressed as follows [27,28]:(3)$$V_{FB}=\frac{\Phi_{M}-\Phi_{S}}{q}-\frac{Q_{OX}-Q_{SS}}{C_{OX}}$$
where $\Phi_{M}$ is the metal work function, $\Phi_{S}$ is the semiconductor work function, $Q_{OX}$ is the oxide charge, and $Q_{SS}$ is the charge between the surface and the interface. Therefore, the C–V curve measured by the EIS structure causes the voltage to shift, owing to the reference electrode potential ($E_{Ref}$), electrolyte ($\chi^{sol}$), and electrolyte and sensing layer interface potential ($\psi_{0}$). The offset voltage is ($E_{Ref}-\psi_{0}+\chi^{sol}$). By substituting the $\frac{\Phi_{M}}{q}$ term in Equation (3) with ($E_{Ref}-\psi_{0}+\chi^{sol}$), the flat band voltage of the EIS structure can be described as follows:(4)$$V_{FB\left(EIS\right)}=E_{Ref}-\psi_{O}+\chi^{sol}-\frac{\Phi_{Si}}{q}-\frac{Q_{OX}+Q_{SS}}{C_{OX}}$$

In the above equation, $\psi_{0}$ is a function of the pH value of the aqueous solution; the other items are approximately independent of the pH value and can thus be considered as constants. Therefore, the C–V curve of the EIS structure undergoes a flat-band shift with the pH value of the aqueous solution. As the pH value of the aqueous solution increases, the curve shifts to the right. Therefore, the voltage shift, $\Delta V_{FB}$, occurs, and the sensitivity can be obtained. Figure 5 shows the capacitance and voltage (C–V) curves of the ZO and MZO sensing film obtained after NH_3_ plasma treatment for 1~6 min in a buffer solution with variable pH. The threshold voltage shift of the MZO film after NH_3_ plasma treatment for 3 min versus the solution’s pH value reveals that the device sensitivity reached as high as 53.82 mV/pH. The sensitivities of the as-deposited ZO samples and samples treated with plasma for 1~6 min were 31.2, 39.06, 47.02, and 39.94 mV/pH, respectively. The sensitivity of the MZO samples was 37.12, 44.02, 53.82, and 40.88 mV/pH, respectively. The ZO and MZO samples treated with plasma for 3 min had the highest sensitivity, as shown in Figure 5. Therefore, the pH sensing film treated with NH_3_ plasma for 3 min achieved excellent linearity and had high sensitivity. This can be attributed to the formation of a greater number of nitride bonds (N-Zn bonds), which increases the number of surface sites [29,30] in the ZO and MZO film surface, owing to the plasma processing. Additionally, the investigation also revealed that the capacitance of the EIS device subjected to NH_3_ plasma treatment for 6 min was smaller compared with the capacitance of the EIS device treated for 3 min. This indicates that prolonged NH_3_ plasma treatment does damage the film’s surface or reduce sensitivity [31]. However, Mg^2+^ ions can easily form because Mg has a low electronegativity (*X* = 1.31). During the doping of Mg^2+^ into the ZO film, the Zn^2+^ ions were replaced by Mg^2+^ ions that could possibly revise the Zn-O bonds and increase the surface density. Therefore, the MZO exhibited the best sensitivity after NH_3_ plasma treatment for 3 min.

We investigated the film’s hysteresis effects; hysteresis is usually found in conductive metal oxides, depending on various factors, like diffused ions, different oxidation states, and internal defects [32]. Besides having prominent sensitivity of the ZO- and MZO-based sensor, reliability is a major issue. Now, the reliability test can be divided into two types—short-term reliability test (hysteresis) and long-term reliability test (drift rate). Figure 6a,b showed that hysteresis was measured at the pH loop of 7→4→7→10→7 for the duration of 25 min. The hysteresis effect comes from the buried surface defects. We calculated the hysteresis voltage from the difference between the initial (pH 7) and terminal (pH 7) voltages in the loops with different pH concentrations [33]. This caused the asymmetric hysteresis behavior of the ZO and MZO NH_3_-plasma-treated sensing film. As can be seen, all samples exhibited asymmetric hysteresis behavior. Figure 6a,b shows the hysteresis voltage of the abovementioned samples. The hysteresis voltages of the as-deposited ZO samples and samples subjected to NH_3_ plasma treatment for 1 min, 3 min, and 6 min were 22.22, 9.27, 5.87, and 7.46 mV, while the hysteresis voltages of the MZO as-deposited samples and samples subjected to NH_3_ plasma treatment for 1 min, 3 min, and 6 min were 17.23, 8.93, 2.52, and 5.99 mV, respectively. Notably, the samples treated for 3 min exhibited the smallest hysteresis deviation. The MZO EIS capacitor treated for 3 min had a smaller hysteresis voltage of 2.52 mV, compared with the other plasma-treated conditions, owing to the low number of crystal defects. Hence, an appropriate treatment for 3 min can improve the hysteresis effect by repairing the bond connections, fill in holes, and eliminate porous structures. In contrast, the samples treated for 1 min and 6 min had a large hysteresis voltage. This indicates highly dense crystal defects that create interior sites that can respond to changes in the chemical composition of the tested solution, which in turn results in large gate voltage variations [34,35]. For long-term reliability, the drift effect of the ZO and MZO sensing film was measured, using the C–V curve, in a pH 7 buffer solution, for 720 min, as shown in Figure 6c,d. The V_out_ reached an approximately steady state after the initial stabilization period. Using linear fitting with V_out_ and an immersion time from 0 to 720 min, the drift coefficient was calculated. The drift rates of the ZO film for the as-deposited samples and samples subjected to NH_3_ plasma treatment for 1 min, 3 min, and 6 min were 13.49, 6.21, 3.67, and 7.57 mV/h, respectively. The draft rates of the MZO as-deposited samples and samples subjected to NH_3_ plasma treatment for 1 min, 3 min, and 6 min were 11.62, 3.79, 1.75, and 6.32 mV/h, respectively. The superior long-term stability of the nitridated ZO and MZO EIS structures is attributed to the repair of the defects in the ZO and MZO film, which resulted from the incorporation of nitrogen. However, the higher drift rate may have been caused by the highly dense crystal defects. Figure 6 shows the hysteresis voltage and drift rate voltage under different NH_3_ plasma conditions; it also shows that the MZO film sample treated for 3 min exhibited the results of lowest hysteresis voltage of 2.52 mV and lowest drift rate of 1.75 mV/h [36,37].

We investigated the capability of the sensing film of the ZO and MZO treated with NH_3_ plasma for 3 min, in a solution with different ions. In recent years, it was considered that the concentration of human blood potassium ion (K^+^) and sodium ion (Na^+^) have been a highly crucial indicator, associated with the estimator of an illness of kidney. To analyze the K^+^ and Na^+^ sensing characteristics, it measured the ZO and MZO film treated with NH_3_ plasma at 3 min. First, we prepared a buffer solution of 5 mM Tris/HCl with pH 7. Subsequently, to make controllable the K^+^ and Na^+^ concentrations within the range of 10^−5^~10^−1^ M, we used a micropipette by injecting into the buffer electrolyte of 1 M NaCl/Tris-HCl and 1 M KCl/Tris-HCl. Next, we selected the control conditions for the sensing film of the ZO and MZO as-deposited samples and samples treated with NH_3_ plasma for 3 min. The calculated pNa and pK sensitivities of the ZO and MZO sensing film are shown in Figure 7. The pNa and pK sensitivities of the sensing film of the ZO and MZO samples treated with NH_3_ plasma for 3 min were 6.07 mV/pNa and 10.05 mV/pK, and 8.03 mV/pNa and 11.26 mV/pK, respectively. Moreover, we compared the different ion sensitivities of the sensing film of the MZO samples treated with NH_3_ plasma for 3 min, as shown in Figure 7b,d. Obviously, the EIS structure incorporating an MZO sensing film subjected to NH_3_ plasma treatment for 3 min was proved to be more responsive to H^+^, as compared with Na^+^ and K^+^ [38,39,40].

This step was to immobilize the enzyme on the surface of the film, using covalent binding technology. The experimental steps are briefly described below. First, we immersed the sample in a 10% hydrogen peroxide (H_2_O_2_) solution for 24 h. Subsequently, the sample was placed in a 9% (3-Aminopropyl) triethoxysilane (APTS) solution, at 40 °C, for four hours, to be silylated. Third, the sample was immersed in 10% glutaraldehyde, a bifunctional group that functions as a cross-linking enzyme and an amine bond in APTS, for one hour. Next, the sensing film was dripped with enzyme and then stored in a refrigerator, at 4 °C. Finally, the unfixed enzyme was washed with phosphate buffer.

The urea hydrolysis reaction catalyzed by urease is as follows [41,42]: (5)$$NH_{2}CONH_{2}+3H_{2}O\overset{\mathrm{urease}}{\to }2NH_{4}^{+}+OH^{-}+HCO_{3}^{-}$$

The hydrogen ions and ammonium ions produced by the enzyme-catalyzed reaction may have been used in the urea detection, leading to changes in the sensor’s analytical output signal. To investigate the urea-sensing properties of the sensing film of the ZO treated with NH_3_ plasma for 3 min and MZO treated with NH_3_ plasma for 3 min on the EIS structure, we prepared a urea solution whose concentration was controlled in the range of 2–32 mM. Then, we selected the control conditions for the sensing film of the ZO and MZO samples treated with NH_3_ plasma for 3 min. The linearity and sensitivity properties of urea are shown in Figure 8c,d respectively. As can be seen, the ZO film treated with plasma for 3 min had a sensitivity of 3.91 mV/mM; the linearity was 94% in the concentration range of 2–32 mM. The MZO film treated with plasma for 3 min had a sensitivity of 8.39 mV/mM; the linearity was 98.67% in the concentration range of 2–32 mM. Compared with the control ZO and MZO sensing film treated with NH_3_ plasma for 3 min for urea detection, the urea-sensing properties of the MZO sensing film treated with NH_3_ plasma for 3 min had higher values compared with the control samples.

We used enzymes attached to the EIS structure as glucose biosensors and made glucose oxidase a bio-component. The biosensor was used to study the glucose concentration measurement through the detection of the pH variation caused by the redox reactions associated with the dissociation of glucose acid. Typically, glucose oxidase hydrolyzes glucose as follows [43,44]:
Glucose + glucose oxidase (OX) → Gluconolactone + glucose oxidase (R)
glucose oxidase (R) + O_2_ → glucose oxidase (OX) + H_2_O_2_
H_2_O_2_ → O_2_ + 2H^+^ + 2e^−^.

To investigate the glucose-sensing properties of the sensing film of the ZO treated with NH_3_ plasma for 3 min and MZO treated with NH_3_ plasma for 3 min, in the EIS structure, we prepared a glucose solution whose concentration was controlled in the range of 2~7 mM. Then, we selected the control conditions for the sensing film of the ZO and MZO treated with NH_3_ plasma for 3 min. The glucose properties (linearity and sensitivity) are shown in Figure 8a,b, respectively. As can be seen, the ZO film treated with plasma for 3 min had a sensitivity of 5.10 mV/mM; the linearity was 89.01% in the concentration range of 2~7 mM. The MZO film treated with NH_3_ plasma for 3 min had a sensitivity of 10.73 mV/mM; the linearity was 97.87% in the concentration range of 2–7 mM. The glucose-sensing properties of the MZO treated with NH_3_ plasma for 3 min were higher compared with the sensing film of the control ZO and MZO treated with NH_3_ plasma for 3 min. Therefore, glucose and urea produce hydrogen ions, hydroxide ions, and ammonium ions through an oxidation-reduction reaction. The sensing membranes were used to accurately measure those values.

Table 1 compares the bio-sensing parameters, such as the urea and glucose sensing sensitivity, for different EIS devices with In_2_TiO_5_ with CF_4_ plasma [35], TbY_x_O_y_ [45], CeO_2_ [46], ZO [47], Sm_2_O_3_ [48], Ti-doped ZO [49], and CeO treated with CF_4_ plasma [50]. According to our previous study on zinc oxide (ZO) [47], the pH-sensing sensitivity of the ZO as-deposited film and the annealed film at 600 °C were 33.15 and 42.54 mV/pH, respectively; and the hysteresis voltages of the above ZO films were 35.10 and 7.37 mV, respectively. By the way, the sensitivity of glucose and urea measurement of the ZO as-deposited film were 3.14 and 1.81 mV/mM, respectively. Furthermore, the EIS device using the MZO film treated with NH_3_ plasma for 3 min exhibited optimum pH sensitivity (53.82 mV/pH), lower hysteresis (2.52 mV) with better glucose and urea sensing performance (10.73 mV/mM; 8.39 mV/mM, respectively). The Mg incorporated into the ZO film resulted in MgO bonding at the ZO sensing film, which improved the bio-sensing characteristics. The NH_3_ incorporated into the MZO film passivated the film surface at the MZO sensing film and thereby improved the sensing characteristics.

## 4. Conclusions

In this research, the Mg-doped zinc oxide was treated with NH_3_ plasma for 3 min, and the film was deposited on a silicon substrate, to fabricate an EIS structure, and used to measure pH buffer solution with better sensitivity, outstanding linearity, lower hysteresis, and excellent drift ratio voltage property. The 3-min-plasma-treated device had the best material and electrical properties, as well as the most satisfactory sensing capability, possibly owing to the reduction of oxygen vacancy and defect passivation caused by nitrogen. Moreover, the MZO sensing film treated with NH_3_ plasma for 3 min was applied as urea and glucose biosensors, and it exhibited better sensitivity and linearity. The NH_3_ plasma treatment after deposition can enrich the crystal structure and repair the dangling bonds at the grain boundaries. In addition, the 3-min-plasma-treated device exhibited the best material and electrical properties; its attributes give it the most ideal sensing capabilities, possibly due to defect passivation by improving the bonding of nitrogen and magnesium. The results obtained by this study revealed that the zinc oxide sensing film in the EIS structure is promising for developing biomedical device applications.

## Figures and Tables

**Figure 1 nanomaterials-10-00583-f001:**
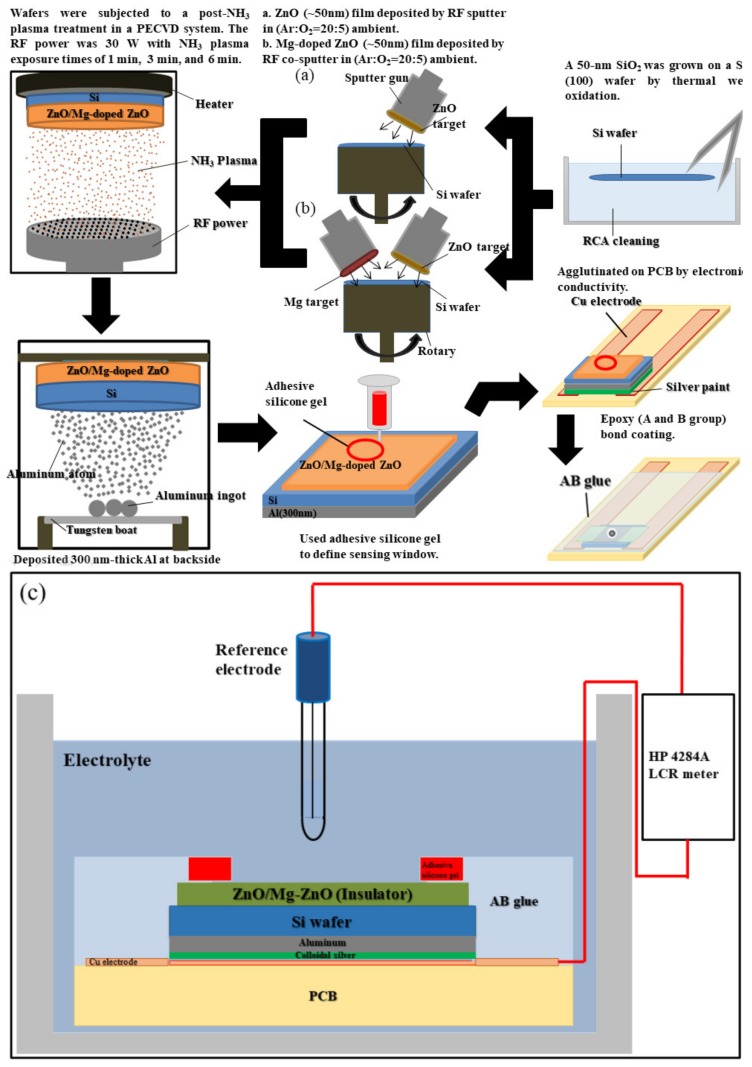
(**a**,**b**) Process flow of ZO and MZO sensing film treated with NH_3_ plasma; (**c**) EIS system.

**Figure 2 nanomaterials-10-00583-f002:**
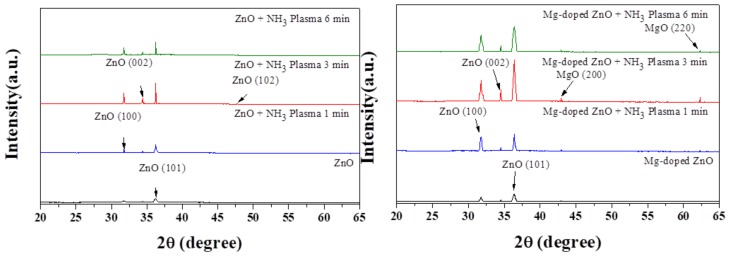
XRD of ZO and MZO film treated with NH_3_ plasma on single crystalline silicon for 1 min, 3 min, and 6 min.

**Figure 3 nanomaterials-10-00583-f003:**
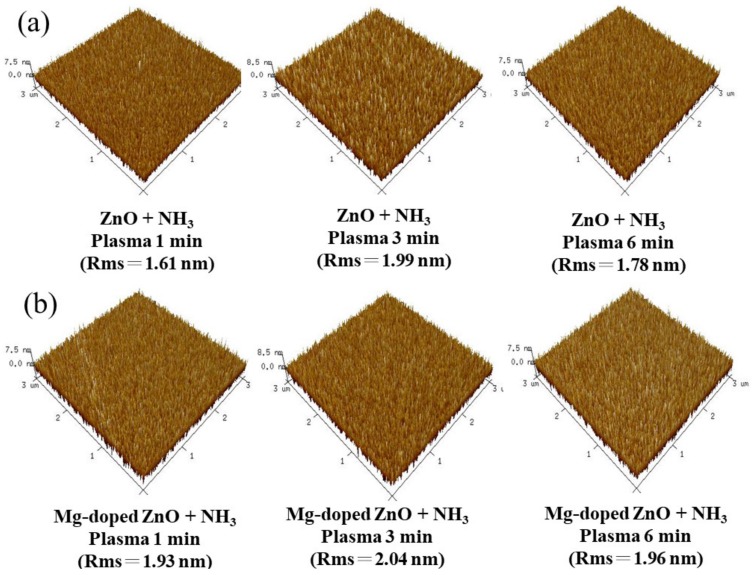
AFM of surface of (**a**) ZO sample treated with NH_3_ plasma for 1 min, 3 min, and 6 min; (**b**) MZO sample treated with NH_3_ plasma for 1 min, 3 min, and 6 min.

**Figure 4 nanomaterials-10-00583-f004:**
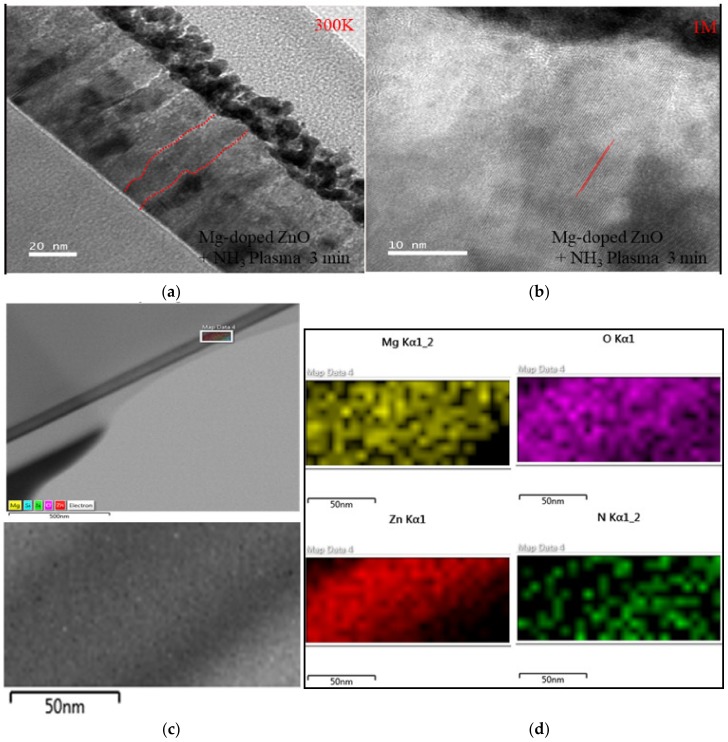
(**a**,**b**) MZO thin film TEM and EDS profiles of samples treated with NH_3_ plasma. (**c**,**d**) EDS analysis of MZO thin film with NH_3_ plasma at 3 min.

**Figure 5 nanomaterials-10-00583-f005:**
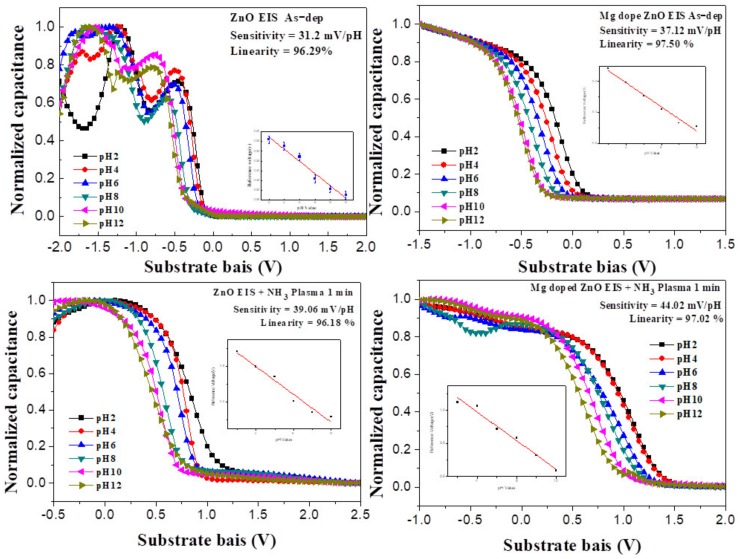
The output characteristics (C–V curve) of the ZO and MZO, with and without NH_3_ plasma treatment, under various factors, were observed in a buffer solution, from pH 2 to pH 12.

**Figure 6 nanomaterials-10-00583-f006:**
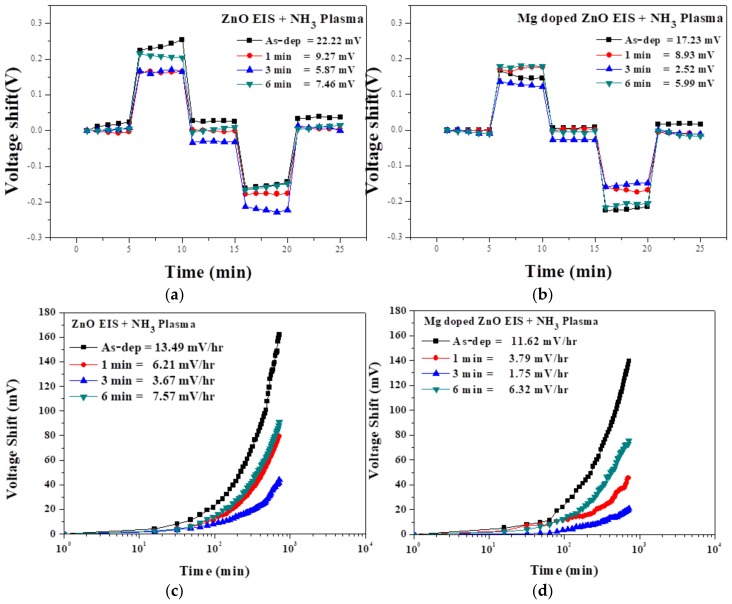
(**a**,**b**) The hysteresis of ZO and MZO sensing film treated with NH_3_ plasma under various conditions during the pH loop of 7, 4, 7, 10, and 7, for a duration of 25 min. (**c**,**d**) Then, the ZO and MZO sensing film treated with NH_3_ plasma under various conditions demonstrates the drift rate to show the long-time reliability of the device in pH 7 buffer solution over a period of 720 min.

**Figure 7 nanomaterials-10-00583-f007:**
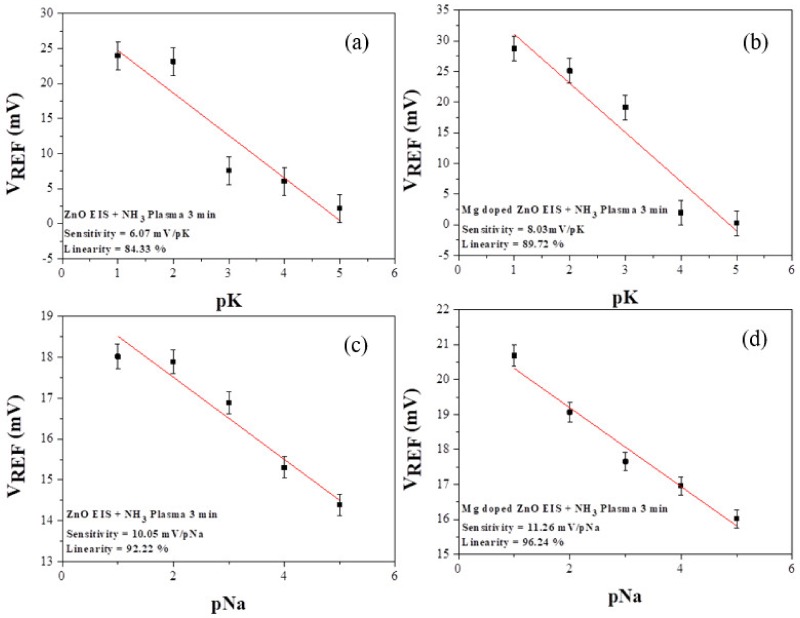
Sensing attribute of (**a**,**b**) K^+^ and (**c**,**d**) Na^+^ by ZO and MZO treated with NH_3_ plasma for 3 min.

**Figure 8 nanomaterials-10-00583-f008:**
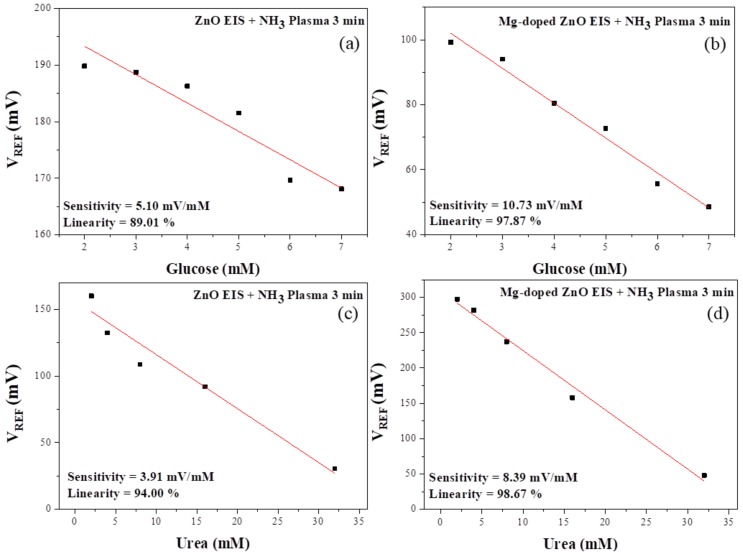
The (**a**,**b**) pGlucose/(**c**,**d**) pUrea responses of EIS structures consisting of enzyme-immobilized ZO and MZO treated with NH_3_ plasma for 3 min, using the covalent bonding method.

**Table 1 nanomaterials-10-00583-t001:** Comparison of obtained parameters for glucose and urea sensing amongst different EIS devices.

Sensing Film	Glucose (2~7 mM)	Urea (5~40 mM)	Reference
TbY_x_O_y_ APTES+GA	4.81 mV/mM	-	[44]
CeO_2_	4.74 mV/mM	2.49 mV/mM	[45]
ZnO	3.14 mV/mM	1.81 mV/mM	[46]
Sm_2_O_3_	-	2.45 mV/mM	[47]
Ti-ZnO	6.42 mV/mM	1.4~3.62 mV/mM	[48]
CeO withCF_4_ plasma	5.83 mV/mM	2.30 mV/mM	[49]
In_2_TiO_5_ with CF_4_ plasma	6.63 mV/mM	2.69 mV/mM	[34]
MZO with NH_3_ plasma	10.73 mV/mM	8.39 mV/mM	This study

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
