# Peer review of "NH3 Plasma-Treated Magnesium Doped Zinc Oxide in Biomedical Sensors with Electrolyte–Insulator–Semiconductor (EIS) Structure for Urea and Glucose Applications"

_nanomaterials, 2020, doi:10.3390/nano10030583_

Round 1
Reviewer 1 Report
This paper investigates a study of a NH3 Plasma-treated Magnesium Doped Zinc Oxide in Biomedical Sensors with Electrolyte-Insulator-Semiconductor (EIS) Structure for Urea and Glucose Applications.
The paper is well organized with proper structure and length. The bibliography is sufficient and well given.
Specifically, the technical terms are explained in detail and the topic of the paper is clear and understandable.
Mathematical models are well written and appropriate refereed.
The presented methodology and the results are clearly communicated, with the necessary background for the readers included in the paper.
The review of the state-of-the-art is sufficient. It includes references to other relevant studies that have been previously proposed for the discovery of relations.
The novel contribution of the paper is highlighted, as well.The conclusion section includes a discussion about the results obtained by this work, but it doesn't demonstrate previous works on the analysis of the same or similar data.
Author Response
The novel contribution of the paper is highlighted, as well. The conclusion section includes a discussion about the results obtained by this work, but it doesn't demonstrate previous works on the analysis of the same or similar data.
Response 1: We have demonstrated the data of our previous works before the conclusion section. According to our previous study in zinc oxide (ZO) [46], the pH sensing sensitivity of the ZO as-deposited film and the annealed film at 600 ℃ were 33.15 mV/pH and 42.54 mV/pH, respectively; and the hysteresis voltages of the above ZO films were 35.10 mV and 7.37 mV, respectively. By the way, the sensitivity of glucose and urea measurement of the ZO as-deposited film were 3.14 mV/mM and 1.81 mV/mM, respectively. Furthermore, the EIS device using the MZO film treated with NH3 plasma for 3 minute exhibited optimum pH sensitivity (53.82 mV/pH), lower hysteresis (2.52 mV) with better glucose and urea sensing performance (10.73 mV/mM; 8.39 mV/mM, respectively). The Mg incorporated into the ZO film resulted in MgO bonding at the ZO sensing film, which improved the bio-sensing characteristics. The NH3 incorporated into the MZO film passivated the film surface at the MZO sensing film and thereby improved the sensing characteristics.

Reviewer 2 Report
Paper quite interesting well presented with only one improving variation to be introduced:the systems seems work both to metal ion (hydrogenfirst) and to molecules of biological interest.The correlation betwee these two quite different cases if any,as it appears reasonable,deserves a deeper analysis fromthe authors.
Author Response
Point 1: Paper quite interesting well presented with only one improving variation to be introduced: the systems seems work both to metal ion (hydrogen first) and to molecules of biological interest. The correlation between these two quite different cases if any, as it appears reasonable, deserves a deeper analysis from the authors.
Response 1: In the bio-measurement, the hydrogen ions and ammonium ions produced by the enzyme-catalyzed reaction may have been used in the urea detection, which led to changes in the sensor’s analytical output signal; and the glucose measurement through the detection of the pH variation caused by the redox reactions associated with the dissociation of glucose acid. Therefore, glucose and urea produce hydrogen ions, hydroxide ions, and ammonium ions through an oxidation-reduction reaction. The sensing membranes were used to accurately measure those values.

Reviewer 3 Report
The manuscript presents a NH3 plasma-treated magnesium doped Zinc oxide for its application as gate oxide in a EIS structure and its sensitivity for potentiometric detection of proton, of alkaline ions and of enzymatic substrates (urea and glucose). The material is well characterized for its crystallinity and its morphology. The influence of the NH3 treatment duration is very important and optimum conditions for the potentiometric detection of pH are found and applied for the detection of other species.
The introduction is foccused on the material itself. A part should be added on the materials used gate oxide in EIS structures and their main features.
The detection of urea and glucose required the grafting of the specific enzyme (urease and GOD). This point should be reported in the abstract.
There are a lot of minor points that should be revised before this manuscript should be considered for publication in Nanomaterials:
- In the introduction, last paragraph, the use oh NH3 plasma for post-treatment was performed for ZO and for MZO. This point is not clear at the beginning of the paragraph.
- In page 3, line 5, it should be "ZO and MZO"
- In §3.1., line 7, it should be "ZO and MZO films"
- In §3.1., lines 9-10, the sentence "This analysis ... experiment" is not understandable.
- "film" is prefered to "membrane" in the whole manuscript
- Page 4, from line 7 (from the bottom), it is a new paraphe and an alinea is required.
- Page 5, last paragraph: in the first sentence, TEM image is not presented in Fig. 2, but in Fig. 4. The same problem all along this paragraph.
- In Fig.4, letters a, b, should be added on the different photos. Figure 4c is not found.
- In §3.2, EIS system is not presented in Fig. 1 and MIS structure not in Fig. 3.6. These figures shoud be Added or this point shoud be revised.
- In page 8, line 5, it should be "ZO samples"
- In page 11, line 1, the first sentence is not correct. Grafting of what?
- In page 11, line 16 the senstence:"The urea properties ..." is not correct.
- There are several synthax errors that should be checked with a native english person's help.
Author Response
There are a lot of minor points that should be revised before this manuscript should be considered for publication in Nanomaterials:
Point 1: In the introduction, last paragraph, the use of NH3 plasma for post-treatment was performed for ZO and for MZO. This point is not clear at the beginning of the paragraph.
Response 1: To this end, as the nitrogen (N) is similar to that of oxygen (O) about ionic radius and acts as a better compensator so that NH3 plasma was used in the post-treatment of sputter film of ZO and MZO.
Point 2: In page 3, line 5, it should be "ZO and MZO"
Response 2: I have modified it according to your suggestion.
After deposition, the ZO and MZO was subjected to post-NH3 plasma treatment inside a plasma-enhanced chemical vapor deposition (PECVD) system with an RF power of 30 W and processing pressure of 500 mTorr for 1 minute, 3 minute, and 6 minute.
Point 3: In §3.1., line 7, it should be "ZO and MZO films"
Response 3: I have modified it according to your suggestion.
The data were compared to information provided by the Joint Committee on Powder Diffraction Standards (JCPDS) to clarify the 2θ value of the lead magnesium and its corresponding lattice planes, and determine the crystal phase of the ZO and MZO film prepared using different coating parameters, as shown in Figure 2.
Point 4: In §3.1., lines 9-10, the sentence "This analysis ... experiment" is not understandable.
Response 4: This analysis confirmed the lattice changes of the sensing film in different conditions in the experiment.
Point 5: "film" is prefered to "membrane" in the whole manuscript
Response 5: I have followed your suggestion to replace the "membrane" with "film" in the revised manuscript.
Point 6: Page 4, from line 7 (from the bottom), it is a new paraphe and an alinea is required.
Response 5: I have modified the format of this paragraph based on your suggestion.
Point 7: Page 5, last paragraph: in the first sentence, TEM image is not presented in Fig. 2, but in Fig. 4. The same problem all along this paragraph.
Response 7: The TEM image (Figure 4) of the MZO composite treated with NH3 plasma for 3 minute has been shown in Figure 4. It is clearly evident that the MZO composites have nanoscale structure. Additionally, ZO exhibited similar columnar structures in the polymerization of nanoparticles. The low-resolution TEM images (LRTEM) (Figure 4a) further confirms this columnar structure and reveal its high density. As shown in Figure 4b, the lattice spacing was approximately 0.15~0.2 nm in the high-resolution TEM image (HRTEM). The uniform composition of the MZO composites treated with NH3 plasma for 3 minute was further measured by carrying out EDS analysis in figure 4c and 4d. Figure 4d shows a representative elemental mapping of the nitrogen, oxygen, zinc, and magnesium in the composite [26].
Point 8: In Fig.4, letters a, b, should be added on the different photos. Figure 4c is not found.
Response 8: I have added a, b, c and d to make figure 4 more understandable.
Figure 4. MZO thin film TEM and EDS profiles of samples treated with NH3 plasma.
Point 9: In §3.2, EIS system is not presented in Fig. 1 and MIS structure not in Fig. 3.6. These figures should be added or this point should be revised.
Response 9: I have drawn a schematic figure about the EIS system in figure 1.
The EIS system consisted of a reference electrode, electrolyte, insulating layer, substrate, and metal electrode, as shown in Figure 1(c). Additionally, in electrochemistry, the EIS system is typically expressed as follows:
Point 10: In page 8, line 5, it should be "ZO samples"
Response 10: I have modified it according to your suggestion.
The sensitivities of the as-deposited of ZO samples and samples treated with plasma for 1~6 minute were 31.2, 39.06, 47.02, and 39.94 mV/pH, respectively.
Point 11: In page 11, line 1, the first sentence is not correct. Grafting of what?
Response 11: This step is to immobilize the enzyme on the surface of the film using covalent binding technology. The experimental steps are briefly described below.
Point 12: In page 11, line 16 the sentence: "The urea properties ..." is not correct.
Response 12: I slightly modified on page 11 line 16 the sentence. Then I have redrawn the data about Figure 8.
The linearity and sensitivity properties of urea are shown in Figure 8 (c-d), respectively.
Point 13: There are several synthax errors that should be checked with a native english person's help.
Response 13: We have followed your suggestion to modify the English descriptions by professional editors.
